# "The impact of the COVID-19 pandemic on research activities: A survey of the largest Italian academic community"

**Massimo Volpe[1] \*, Massimo Ralli[2], Andrea Isidori[3]**

**1** Department of Clinical and Molecular Medicine, Faculty of Medicine and Psychology, Sapienza University of Rome, Rome, Italy, **2** Department of Sense Organs, Faculty of Medicine, Sapienza University of Rome, Rome, Italy, **3** Department of Experimental Medicine, Faculty of Medicine, Sapienza University of Rome, Rome, Italy

\* massimo.volpe@uniroma1.it

**Data Availability Statement:** All relevant data are within the manuscript and its Supporting Information files.

**Funding:** The author(s) received no specific funding for this work.

## Abstract

### Purpose

The aim of the present work is to explore the impact of the COVID-19 pandemic on research activities in a vast multidisciplinary academic community to identify the most critical issues.

### Method

To this purpose we planned a survey addressed to the entire academic research staff at "*Sapienza*" University of Rome, which represents the largest Italian academic community. A questionnaire consisting of both open and closed-ended questions was delivered to 4118 individuals in April 2021.

### Results

A total of 544 responses were collected. All academic roles were sufficiently represented in the study cohort. The median number of critical issues experienced by academic research staff was three. Among these, the three most frequently reported were related to: "Access to libraries / laboratories / research sites" (21.9%), "Limitation to stay abroad / study / research periods" (17.6%), "Progress of experimental work" (14.7%), with variable prevalence according to academic position and gender. Older subjects reported issues with "Projects' financial reporting" and "Expiration of acquired consumable material more frequently". The most common critical aspects reported in relation to the economic burden were: being "Unable to allocate funds" (31.4%), a "Reduction in clinical and scientific activity" (26.3%) and experiencing "Increased expenses (comprising private costs)" (21.2%) with no differences between genders. Researchers in Applied Sciences and Natural Sciences reported a higher frequency of problems in clinical and scientific activities, whereas increased expenses were reported also by researchers operating in the Humanities field. As a possible solution aimed at improving these issues, most subjects, especially those aged >45 years, indicated "Economic aid" (22.6%), "Reduction in bureaucracy" (19.9%) or "Enhancement of

**Competing interests:** The authors have declared that no competing interests exist.

the scientific and clinical activities", whereas those aged ≤45 years felt that an increased duration and better access to PhD programs were to be prioritized.

## Conclusion

Our findings highlight the most critical issues related to research activities during the COVID-19 pandemic in a large academic community. The information achieved may be useful to identify researchers' needs and to design appropriate policies aimed at preparing research institutions for unexpected catastrophic events and limiting the negative impact on academic research activities.

## Introduction

Since early 2020, the outbreak of the COVID-19 pandemic has disrupted not only societal and individual lifestyles around the world but has also heavily affected and impaired scientific research activities. Beside causing an enormous impact on healthcare systems, the pandemic outbreak caused by SARS-CoV-2 has also heavily limited research programs in all academic fields by interfering with key steps of research activities at multiple levels [1]. A recent publication by Myers et al. reported that since the onset of the COVID-19 pandemic and the consequent disruption of research activities, faculties have lost, on average, 24% of their research productivity [2]. Laboratory-based scientists have been the most affected, as they lost up to 40% of their productivity [3]. Indeed, many research institutions and universities from different countries have severely reduced on-site academic activities [3]. The COVID-19 pandemic affected scientists in all fields [4], particularly in scientific fields that require laboratory resources, living animals, and time-sensitive experiments [5]. Within these settings, healthcare research has experienced the greatest impact. In the UK, preliminary estimates suggest that over 1,500 academic trainees have been re-assigned exclusively to clinical activities during the pandemic with no spare time for research activities [6]. Among the multiple consequences of the COVID-19 pandemic, closure of laboratories and research institutes, staff shortages, supply chain disruptions for reagents and laboratory materials as well as funding deficits have represented significant, often insurmountable, hardships for the "physiological" flow of research [7–10]. Moreover, the COVID-19 pandemic has adversely affected clinical trials, which were often delayed or deferred [3]. An impressive shift of research activities towards those focused on COVID-19 was also recorded [1]. In terms of publishing trends, some have reported a decrease in research quality and accuracy, publication of incomplete or interrupted scientific studies, and a reduction in the space given to important non-COVID work [11, 12]. In addition to the impact on scientific publication, the pandemic also affected gender disparities. Studies suggested that the temporary shutdown of social relations during the pandemic, the increased caregiving responsibility for family, and the reduction of time available to work had an especially significant effect on women, who are often more involved in domestic responsibilities and childbearing duties [13, 14]. Studies suggest that the pandemic exacerbated existing gender disparities for career progress [15]. Indeed, numerous studies focused on disproportionate changes in publications between men and women during the pandemic [16]. Beside a general halt to research activities, emphasis has been placed on the effects on Ph.D. students, postdoctoral fellows and junior faculty, for whom the toll of lost research-time may be greater, as they are expected to produce their academic work and achieve research results within the limited timespan of their fellowships [17]. Consequently, early-career researchers (ECR) have

experienced repercussions on careers and job opportunities, especially Ph.D. students and younger researchers [17–19]. On the contrary, some potentially positive "side effects", such as an acceleration of digitalization processes, adoption of remote work and remote connections, and a higher productivity in specific fields, such as artificial intelligence [20] ought to be mentioned. The dramatic period experienced by the scientific community during the COVID-19 pandemic has revealed the most fragile aspects of academic organizations and research activities, and the lessons learned should promote actions to respond and be prepared to preserve scientific activities during challenging times and emergencies. A comprehensive analysis of the multifaceted impacts of the COVID-19 pandemic on different areas of research within the realms of academia may facilitate the development of strategies to mitigate the consequences of the COVID-19 pandemic or other global catastrophic events. Therefore, to explore the impact of the COVID-19 pandemic on research activities and in the attempt to identify the most critical issues, we performed a survey within our academic community based in the *Sapienza University of Rome*, Italy. To our knowledge, our current work is the first study in Italy that attempts to identify the impact of the COVID-19 pandemic on research activities of an entire academic community involving all areas and disciplines.

## Methods

*Sapienza University of Rome* is the largest European university by student number with ~117.000 students. At the time of the survey, the Sapienza community included 3300 academics, 845 fixed-term researchers, and 2874 Ph.D. students, all involved in academic research activities. For its size and characteristics, Sapienza University is highly representative of a large multidisciplinary academic population, spanning different ages, genders, and career stages and acting as an international environment that boosts international relationships and cooperation (with about 9% of international students) [21]. For the scope of our research, the Governance Research Team of the University designed a cross-sectional study through the use of an online survey, consisting of both open-ended and closed-ended questions. The study aimed to explore the differences in impact of the pandemic on different academic subgroups on whom, based on the available literature, a higher toll from the pandemic might be expected. The closed-ended questions were related to the research field, the main factors influencing research activity during the COVID-19 pandemic, and the consequent financial impact of research activity limitations. The open-ended questions addressed the specific effects of the financial issues on research activity and required suggestions and feedback about emerging post-pandemic research priorities. The full questionnaire is attached as *S1 Appendix*. This questionnaire, proposed in Italian to participants, comprised the following items: age, sex, academic position, department, faculty, research type, main critical issues experienced during the pandemic, additional issues, absence/presence of economic impact (and its entity), reasons underlying the economic burden (if any), and suggestions to improve academic research and performance. Sociodemographic and income data were also collected. The questionnaire was developed using Google Forms (Google LLC, USA). Each individual participating in this survey has given explicit informed consent to the use of data. The study was conducted in accordance with the Declaration of Helsinki. All data were collected in an anonymous and aggregated form. The protocol for the current survey has been approved by the Sapienza Ethics Committee for Transdisciplinary Research (CERT) (approval no. CERT protocol ID 64/2023). The questionnaire was sent via email in April 2021 to all faculty members (full, associate, and assistant professors), Ph.D. students and medical residents of Sapienza University, which counts 4118 individual email addresses. The academic position was classified into the following categories: "Ph.D. student", "Fixed-term researcher", "Full-term researcher", "Associate

professor" and "Full professor". We categorized all respondents according to their Department and Faculty into academic fields. Respondents were asked closed-ended and open-ended questions, categorized into broad groups (See **S1 Appendix**) Responses were exported in an Excel database (Microsoft Corp, USA) and analyzed using statistical software SPSS Statistics for Windows (version 27, IBM Corp). Data distribution was visually inspected by analyzing the respective histograms and normality plots. Data are presented as counts, percentages (%), means and standard deviations (SD), and medians with 25–75% interquartile ranges (IQR), as appropriate. Analyses were conducted employing $\chi_2$ tests for categorical variables, ANOVA, linear regression and logistic regression analyses were used for X variables. Given the exploratory nature of the study focused on an objective, rather than on pre-specified hypotheses, we chose not to conduct post-hoc tests or to adjust for multiple comparisons [22–24]. A robust approach using bootstrapping for 2000 samples was employed to account for non-normal data distribution, and bias-corrected accelerated (BCa) 95% confidence intervals were calculated and reported. The linear regression analyses used Stein's formula for adjusted R squared (Adj $R^2$) to evaluate how well the models cross-validate across different samples of data from the same population, while for logistic regression analyses, we reported $R^2$ values according to Nagelkerke ($R^2_N$) [25]. The level of statistical significance was set at a P-value <0.05. All statistical computations were conducted with the IBM SPSS Statistics for Windows (version 27, IBM Corp.). We adopted the consensus-based checklist for reporting of survey studies (CROSS) and present the completed survey in Supplementary material [26]. The full anonymized dataset underlying the findings described in this manuscript has been uploaded in Figshare (Figshare LLC, USA), accessible at the following link: https://doi.org/10.6084/m9.figshare.23551743.v1.

## Results

Five hundred and forty-four individuals responded to the questionnaire, accounting for approximately 13.2% of the total invited cohort. The descriptive characteristics of the research cohort are presented in Table 1 according to the academic position, and divided on the basis of the academic field, research type and economic burden experienced. Overall, 52.2% of the respondents were female and the median age was 48 years (35–57); all academic roles were adequately represented in the research cohort, thus reflecting a substantially well-balanced sample of Ph.D. students (22.2%), Fixed-term researchers (13.1%), Full-term researchers (13.1%), Associate professors (36.0%) and Full professors (15.6%).

### Research-related issues experienced

The respondents were asked to state the main issue affecting academic research experienced during the pandemic and to list all issues encountered. The description of the main and the general research-related issues is presented in Table 2.

The median number of issues experienced was 3 (2–4). Among the main issues, the five most frequently encountered were related to: "Access to libraries / laboratories / research site" (21.9%), "Limitation to abroad stay / study / research periods" (17.6%), "Advancement of experimental work" (14.7%), "Scientific production" (11.6%) and "Impact on Family organization" (7.9%). Interestingly, considering all research issues experienced, these five also represented the five most frequently encountered, albeit with a different frequency: "Limitation to abroad stay / study / research periods" (51.3%), "Access to libraries / laboratories / research sites" (48.5%), "Scientific production" (38.1%), "Advancement of experimental work" (33.6%) and "Family organization" (26.5%). Table 3 presents the frequency of research-related issues encountered according to their academic position. Specifically, "Access to libraries /

**Table 1. Description of the research cohort.**

| | Total | PhD student | Fixed-term researcher | Full-term researcher | Associate professor | Full professor |
|---|---|---|---|---|---|---|
| n (%) | 544 | 121 (22,2) | 71 (13.1) | 71 (13.1) | 196 (36.0) | 85 (15.6) |
| Females (%) | 52.2 | 62.8 | 62 | 54.9 | 46.4 | 40 |
| Age (yrs, [25–75% IQR]) | 48 [35–57] | 28 [27–31] | 37.5 [34–44] | 49 [43–56] | 53 [47–59] | 59 [55–63] |
| **Academic field (%)** | | | | | | |
| Professions and app. sciences | 33.1 | 32.2 | 36.6 | 42.3 | 30.6 | 29.4 |
| Humanities | 22.8 | 24 | 15.5 | 12.7 | 22.4 | 17.6 |
| Natural sciences | 18.4 | 18.2 | 19.7 | 16.9 | 20.4 | 14.1 |
| Social sciences | 17.6 | 17.4 | 22.5 | 25.4 | 20.4 | 20 |
| Formal sciences | 8.1 | 8.3 | 5.6 | 2.8 | 6.1 | 18.8 |
| **Research type (%)** | | | | | | |
| Theoretical | 30.1 | 39.7 | 26.8 | 23.9 | 26 | 34.1 |
| Basic | 20.6 | 11.6 | 23.9 | 19.7 | 28.1 | 14.1 |
| Technical/Applied | 17.5 | 19 | 21.1 | 11.3 | 15.8 | 21.2 |
| Field | 13.2 | 10.7 | 9.9 | 16.9 | 15.3 | 11.8 |
| Clinical | 11.2 | 13.2 | 8.5 | 18.3 | 8.7 | 10.6 |
| Biological | 7.4 | 5.8 | 9.9 | 9.9 | 6.1 | 8.2 |
| **Economic burden (%)** | | | | | | |
| None | 68 | 67.8 | 76.1 | 69 | 68.9 | 58.8 |
| <10 000 euro | 11 | 14 | 8.5 | 14.1 | 10.7 | 7.1 |
| 10 000–30 000 euro | 7.9 | 1.7 | 2.8 | 5.6 | 10.2 | 17.6 |
| 30 000–60 000 euro | 1.8 | / | 2.8 | 1.4 | 1.5 | 4.7 |
| 60 000–100 000 euro | 0.6 | / | 1.4 | 1.4 | 0.5 | / |
| >100 000 euro | 0.4 | / | / | / | / | 2.4 |

Note: academic fields were categorized into the 5 major scientific branches, **Formal sciences** (mathematics, logic, statistics, computer science, etc.), **Natural sciences** (physics, chemistry, biology, earth sciences, etc.) and **Social sciences** (archaeology, geography, anthropology, psychology, sociology, economics, political science, cultural/ethnic studies, gender studies, etc.), and **Humanities** (literature, philosophy, theology, linguistics, human history, arts, etc.) and **Professions and applied sciences** (medicine, law, agriculture, education, physical performance, consumer science, journalism, military sciences, engineering, architecture and design, transportation, business, etc.).

laboratories / research sites" showed a significant difference in frequency, being highest among Ph.D. students (66.9%) and Fixed-term researchers (50.7%), whereas a "Limitation to abroad stay / study / research periods" was reported especially by Full professors (62.4%) and Ph.D. students (62%) and "Family organization" was frequent among Associate professors (34.2%) and Full-term researchers (28.2%). On the other hand, no significant differences were observed regarding the frequency distribution of "Advancement of experimental work" and "Scientific production".

We further analyzed the impact of gender, age and academic field on the reported research issues. The full data are reported in Table 4.

No significant differences were found between males and females in the reported issues, except for "Patients" enrolment / access" which was more frequent among females. We divided the research cohort according to age into two groups, ≤45 (n = 241) vs >45 years of age (n = 303). The younger cohort experienced significantly more difficulties regarding "Access to libraries / laboratories / research sites", "Start of experimental work / obtainment of authorizations" and "Impact on family organization". In contrast, the older cohort had higher rates of issues with "Projects' financial reporting" and "Expiry of acquired consumable material". Most research issues significantly differed in frequency among academic fields; complete results are

**Table 2. Description of the main and overall research-related issues experienced.**

| | | |
|---|---|---|
| Median number of issues experienced (IQR 25–75%) | 3 [2–4] | |
| List of issues experienced (n, %) | Main | Overall |
| Access to libraries / laboratories / research sites | 119 (21.9) | 264 (48.5) |
| Limitation to abroad stay /study / research periods | 96 (17.6) | 279 (51.3) |
| Advancement of experimental work | 80 (14.7) | 183 (33.6) |
| Scientific production | 63 (11.6) | 207 (38.1) |
| Impact on Family organization | 43 (7.9) | 144 (26.5) |
| Patients' enrollment / access | 35 (6.4) | 83 (15.3) |
| Sources retrieval | 35 (6.4) | 130 (23.9) |
| Start of experimental work / obtainment of authorizations | 24 (4.4) | 87 (16) |
| Transport logistics | 14 (2.6) | 97 (17.8) |
| Limitation to supply of services (maintenance, etc.) | 14 (2.6) | 73 (13.4) |
| Contract expiry of personnel specifically recruited for research | 8 (1.5) | 50 (9.2) |
| Projects' financial reporting | 6 (1.1) | 63 (11.6) |
| Access to calls with deadlines (awards, grants, etc.) | 5 (0.9) | 38 (7) |
| Expiry of acquired consumable material | 2 (0.4) | 34 (6.3) |

shown in Table 4. A multiple regression analysis of the number of research-related issues experienced, considering respondents' age, gender, and academic position, found a younger age as the only significant 228 independent predictors of a larger sum of overall issues reported (Table 5).

## Experience and causes of economic burden

Overall, 117 respondents (21.7%) reported the presence of an economic burden deriving from the pandemic, with no significant differences according to gender, age group or academic field. In half of the cases reporting an economic burden, it was below 10 000 euros. Complete results are presented in Table 6.

The three most frequently reported reasons underlying an economic burden were: being "Unable to allocate funds" (31.4%), a "Reduction in clinical and scientific activity" (26.3%) and experiencing "Increased expenses (comprising private costs)" (21.2%). No specific differences were reported between males and females or between younger and older respondents. However, those in the fields of Healthcare Professions and Applied Sciences and Natural Sciences reported a higher frequency of problems in clinical and scientific activities (39.5% and 39.1%, respectively), while increased 252(also personal) expenses were reported by those in the Humanities field (40%). When analyzed according to academic role, Ph.D. students reported "Increased expenses (comprising private costs)" more frequently when compared to other groups (38.9%), and they were the only category to mention "Closure of archives and libraries" as problematic (11.1%). Full 256 results are reported in Table 7.

In a logistic regression analysis on the presence of economic burden, comprising age, gender and academic role, only the position of Full professor was found to be a significant independent predictor of the presence of (any) economic burden (Table 8).

## Suggestions to support academic research

Respondents were asked to provide suggestions on how to improve academic research status following the pandemic; answers were categorized into nine broad groups and the results are reported in Table 6. The three most frequent suggestions were the need for "Economic aid"

**Table 3. Research issues experienced based on academic position.**

| | PhD students | Fixed-term researchers | Full-term researchers | Associate professors | Full professors | p |
|---|---|---|---|---|---|---|
| **n** | 121 | 71 | 71 | 196 | 85 | |
| **List of overall issues experienced [n (%)]** | | | | | | |
| Access to libraries / laboratories / research site | 81 (66.9) | 36 (50.7) | 28 (39.4) | 78 (39.8) | 41 (48.2) | **<0.0001** |
| Limitation to abroad stay /study / research periods | 75 (62) | 33 (46.5) | 20 (28.2) | 98 (50) | 53 (62.4) | **<0.0001** |
| Advancement of experimental work | 42 (34,7) | 24 (33.8) | 23 (32.4) | 65 (33.2) | 29 (34.1) | 0.99 |
| Scientific production | 44 (36.4) | 31 (43.7) | 30 (42.3) | 80 (40.8) | 22 (25.9) | 0.11 |
| Family organization | 29 (24) | 17 (23.9) | 20 (28.2) | 68 (34.2) | 11 (12.9) | **0.005** |
| Patients' enrolment / access | 16 (13.2) | 13 (18.3) | 15 (21.1) | 27 (13.8) | 12 (14.1) | 0.53 |
| Sources retrieval | 43 (35.5) | 20 (28.2) | 13 (18.3) | 41 (20.9) | 13 (15.3) | **0.004** |
| Start of experimental work / obtainment of authorizations | 31 (25.6) | 13 (18.3) | 9 (12.7) | 26 (13.3) | 8 (9.4) | **0.011** |
| Transport logistics | 24 (19.8) | 15 (21.1) | 14 (19.7) | 33 (16.8) | 11 (12.9) | 0.64 |
| Limitation to supply of services (maintenance, etc.) | 12 (9.9) | 8 (11.3) | 10 (14.1) | 26 (13.3) | 17 (20) | 0.31 |
| Contract expiry of personnel specifically recruited for research | / | 9 (12.7) | 6 (8.5) | 19 (9.7) | 16 (18.8) | **<0.001** |
| Projects' financial reporting | 4 (3.3) | 12 (16.9) | 6 (8.5) | 28 (14.3) | 13 (15.3) | **0.01** |
| Access to calls with deadlines (awards, grants, etc.) | 7 (5.8) | 8 (11.3) | 5 (7) | 12 (6.1) | 6 (7.1) | 0.65 |
| Expiry of acquired consumable material | 3 (2.5) | 3 (4.2) | 5 (7) | 17 (8.7) | 6 (7.1) | 0.24 |

Note: significant *P*-values are presented in bold.

(22.6%), asking for a "Reduction in bureaucracy" (19.9%), or the "Enhancement of the scientific and clinical activities" (19.2%).No significant differences were reported according to gender. Those ≤45 years of age felt that increasing the length and improving the ease of access of the Ph.D. program (22.9% vs 3.4%) was necessary, together with re-opening archives and libraries (10.2% vs. 2.8%), while those aged >45 years suggested a reduction in bureaucracy more frequently (24.6% vs 12.7%). Among the different academic fields, Natural Sciences requested more "Economic aid" while Humanities favored a larger adoption of "Digitalization" and a prompt "Re-opening of archives and libraries". A "Reduction in bureaucracy" was advocated especially by Associate and Full professors (29.4% and 22.8%, respectively) and, unsurprisingly, PhD students suggested "Increasing the PhD length and ease of access" (31.3%). The full results are reported in Table 6.

## Discussion

The global storm that hit human activities during the COVID-19 pandemic has heavily and broadly affected academic research activities. Although several previous reports have proposed analyses mostly focused on the hardships in specific fields, such as science and medicine, other important aspects have been substantially neglected. A comprehensive exploratory survey, encompassing research difficulties in different disciplines and at different career stages, was lacking in Europe. For this reason, we investigated a representative sample of the academic population of the *Sapienza University of Rome* with a web-based questionnaire. This study presents a unique, qualitative data-driven representation of the impact of COVID-19 on academic research activities in Italy. The results of our survey highlight several critical issues that influenced research activities and could contribute to develop future strategies to limit the consequences of pandemics or other catastrophic events in the future, both in Italy and in other

**Table 4. Research issues experienced based on sex, age (≤45/>45 years) and scientific field.**

| | Males | Females | p | ≤45 yrs | >45yrs | p | PAS | Hum | NS | Soc. Sci. | For. Sci. | p |
|---|---|---|---|---|---|---|---|---|---|---|---|---|
| **n** | **260** | **284** | | **241** | **303** | | **180** | **108** | **100** | **112** | **44** | |
| **List of overall issues experienced (%)** | | | | | | | | | | | | |
| Access to libraries / laboratories / research site | 48.1 | 48.9 | 0.84 | 55.2 | 43.2 | **0.01** | 38.3 | 88.9 | 35 | 49.1 | 20.5 | **<0.001** |
| Limitation to abroad stay /study / research periods | 49.2 | 53.2 | 0.36 | 53.1 | 49.8 | 0.45 | 41.1 | 69.4 | 50 | 49.1 | 56.8 | **<0.001** |
| Advancement of experimental work | 33.5 | 33.8 | 0.91 | 32.4 | 34.7 | 0.58 | 43.3 | 13 | 55 | 20.5 | 29.5 | **<0.001** |
| Start of experimental work / obtainment of authorizations | 16.9 | 15.1 | 0.55 | 20.3 | 12.5 | **0.01** | 22.8 | 5.6 | 20 | 16.1 | 4.5 | **<0.001** |
| Scientific production | 39.2 | 37 | 0.60 | 41.9 | 35 | 0.10 | 37.8 | 34.3 | 40 | 33 | 56.8 | 0.07 |
| Family organization | 25.8 | 27.1 | 0.76 | 31.5 | 22.4 | **0.02** | 20 | 26.9 | 23 | 33 | 43.2 | **0.01** |
| Patients' enrollment / access | 10.8 | 19.4 | **0.01** | 14.5 | 15.8 | 0.67 | 29.4 | 1.9 | 6 | 19.6 | / | **<0.001** |
| Sources retrieval | 23.8 | 23.9 | 0.98 | 27.4 | 21.1 | 0.09 | 14.4 | 49.1 | 5 | 38.4 | 6.8 | **<0.001** |
| Transport logistics | 20.4 | 15.5 | 0.13 | 20.7 | 15.5 | 0.11 | 15 | 19.4 | 29 | 11.6 | 15.9 | **0.012** |
| Limitation to supply of services (maintenance, etc.) | 11.9 | 14.8 | 0.35 | 10.8 | 15.5 | 0.11 | 12.2 | 5.6 | 30 | 8.9 | 11.4 | **<0.001** |
| Contract expiry of personnel specifically recruited for research | 7.7 | 10.6 | 0.26 | 7.1 | 10.9 | 0.12 | 8.9 | 5.6 | 14 | 10.7 | 4.5 | 0.20 |
| Projects' financial reporting | 11.5 | 11.6 | 0.99 | 8.3 | 14.2 | **0.03** | 10 | 10.2 | 11 | 17 | 9.1 | 0.39 |
| Access to calls with deadlines (awards, grants, etc.) | 5.8 | 8.1 | 0.30 | 7.9 | 6.3 | 0.46 | 6.1 | 6.5 | 5 | 9.8 | 9.1 | 0.64 |
| Expiry of acquired consumable material | 5.8 | 6.7 | 0.68 | 3.7 | 8.3 | **0.03** | 9.4 | 1.9 | 7 | 4.5 | 6.8 | 0.12 |

Note: significant *P*-values are presented in bold. Academic fields were categorised into the 5 major scientific branches, **Formal sciences** (mathematics, logic, statistics, computer science, etc.), **Natural sciences** (physics, chemistry, biology, earth sciences, etc.) and **Social sciences** (archaeology, geography, anthropology, psychology, sociology, economics, political science, cultural/ethnic studies, gender studies, etc.), and **Humanities** (literature, philosophy, theology, linguistics, human history, arts, etc.) and **Professions and applied sciences** (medicine, law, agriculture, education, physical performance, consumer science, journalism, military sciences, engineering, architecture and design, transportation, business, etc.).

Abbreviations: **PAS** = Professions and Applied Sciences; **Hum** = Humanities; **NS** = Natural Sciences; **Soc. Sci.** = Social Sciences; **For. Sci.** = Formal sciences.

countries with similar academic systems. A first conclusion that can be drawn from the analysis of our survey is that most of the pre-existing drawbacks and barriers to academic activity have been amplified during the COVID-19 pandemic. In fact, the responses to the questionnaires reflected several weaknesses exacerbated by the pandemic which may require attention and preventive measures. Respondents reported a median of 3 research-related critical issues, with no significant differences among gender, age groups, academic position, research field or type. However, when accounting for age, gender and academic position, younger age was independently associated with a higher number of experienced critical issues (Table 5). When asked about the main issue impacting research, difficulties concerning impaired access to

**Table 5. Linear model of predictors of the number of research-related issues experienced.**

| | b | SE | β | p |
|---|---|---|---|---|
| **Research-related issues number** | | | | |
| Constant | 3.82 (3.31, 4.42) | 0.30 | | <0.001 |
| Age, years | -0.022 (-0.04, -0.01) | 0.01 | -0.21 | **0.008** |
| Sex, male | 0.11 (-0.12, 0.34) | 0.12 | 0.04 | 0.35 |
| Academic position | -0.05 (-0.04, -0.15) | 0.05 | 0.09 | 0.25 |

*Note*: 95% bias corrected accelerated (BCa) confidence intervals, SE and *P* values from 2000 bootstrapped samples are reported in parentheses. Significant *P*-values are presented in bold. *P* for the model = 0.006; Adj $R^2$ = 0.01.

Abbreviations: **β** = standardized coefficient; **b** = unstandardized coefficient; **SE** = standard error.

**Table 6. Presence of economic burden, causes and suggestions to improve research according to sex, age range (45/>45 years) and scientific field.**

| | Total | Males | Females | p | ≤45yrs | >45yrs | p | PAS | Hum | NS | Soc. Sci. | For. Sci. | p |
|---|---|---|---|---|---|---|---|---|---|---|---|---|---|
| **n** | **544** | **260** | **284** | | **241** | **303** | | | 180 124 100 / 25 22.6 26 | | **96** | **44** | |
| **Presence of any economic burden (%)** | **21.7** | **25** | **18.7** | **0.07** | **19.5** | **23.4** | **0.27** | | | | **13.5** | **13.6** | **0.13** |
| **List of reasons for economic burden (%)** | | | | | | | | | | | | | |
| Unable to allocate funds | 31.4 | 27.7 | 35.8 | 0.19 | 28.6 | 38.8 | 0.28 | 30.2 | 28 | 47.8 | 33.3 | 50 | 0.52 |
| Reduction in clinical and scientific activity | 26.3 | 30.8 | 20.8 | 0.19 | 19 | 35.8 | 0.06 | 39.5 | 8 | 39.1 | 16.7 | 33.3 | **0.046** |
| Increased expenses (comprising private costs) | 21.2 | 21.5 | 20.8 | 0.99 | 31 | 17.9 | 0.12 | 27.9 | 40 | / | 25 | / | **0.01** |
| Interrupted mobility programs | 5.9 | 6.2 | 5.7 | 0.95 | 9.5 | 4.5 | 0.30 | 2.3 | 8 | 4.3 | 25 | / | 0.07 |
| Reduction in the number of calls and grants available | 2.5 | 3.1 | 1.9 | 0.71 | 2.4 | 3 | 0.85 | / | 8 | / | / | 16.7 | 0.06 |
| Closure of archives and libraries | 1.7 | 1.5 | 1.9 | 0.86 | 4.8 | / | 0.07 | / | 8 | / | / | / | 0.14 |
| Lack of personnel | 1.7 | 1.5 | 1.9 | 0.86 | 4.8 | / | 0.07 | / | / | 8.7 | / | / | 0.11 |
| **Suggestions to improve academic research (%)** | | | | | | | | | | | | | |
| Economic aid | 22.6 | 19.4 | 25.2 | 0.24 | 18.6 | 25.1 | 0.19 | 23 | 13.2 | 38.5 | 24 | 10.5 | **0.01** |
| Reduction in bureaucracy | 19.9 | 23.9 | 16.6 | 0.12 | 12.7 | 24.6 | **0.012** | 19 | 15.8 | 21.2 | 18 | 42.1 | 0.14 |
| Enhancement of the scientific and clinical activities | 19.2 | 20.1 | 18.4 | 0.71 | 18.6 | 19.6 | 0.85 | 25 | 11.8 | 13.5 | 24 | 21.1 | 0.15 |
| Increasing the PhD length and ease of access | 11.1 | 8.2 | 13.5 | 0.15 | 22.9 | 3.4 | **<0.001** | 14 | 6.6 | 11.5 | 14 | 5.3 | 0.48 |
| Digitalization | 9.8 | 11.9 | 8 | 0.25 | 8.5 | 10.6 | 0.54 | 7 | 22.4 | 3.8 | 6 | / | **0.001** |
| Resumption of presence activities | 6.4 | 6.7 | 6.1 | 0.84 | 3.4 | 8.4 | 0.09 | 4 | 6.6 | 7.7 | 6 | 15.8 | 0.42 |
| Re-opening of archives and libraries | 5.7 | 5.2 | 6.1 | 0.74 | 10.2 | 2.8 | **0.008** | 2 | 15.8 | / | 6 | / | **<0.001** |
| Increasing the personnel | 3 | 2.2 | 3.7 | 0.47 | 2.5 | 3.4 | 0.69 | 5 | 3.9 | 1.9 | / | / | 0.42 |
| Enhancement of abroad mobility programs | 2.4 | 2.2 | 2.5 | 0.90 | 2.5 | 2.2 | 0.86 | 1 | 3.9 | 1.9 | 2 | 5.3 | 0.66 |

Note: significant *P*-values are presented in bold. Academic fields were categorized into the 5 major scientific branches, **Formal sciences** (mathematics, logic, statistics, computer science, etc.), **Natural sciences** (physics, chemistry, biology, earth sciences, etc.) and **Social sciences** (archaeology, geography, anthropology, psychology, sociology, economics, political science, cultural/ethnic studies, gender studies, etc.), and **Humanities** (literature, philosophy, theology, linguistics, human history, arts, etc.) and **Professions and applied sciences** (medicine, law, agriculture, education, physical performance, consumer science, journalism, military sciences, engineering, architecture and design, transportation, business, etc.). Abbreviations: *PAS* = Professions and Applied Sciences; *Hum* = Humanities; *NS* = Natural Sciences; *Soc. Sci.* = Social Sciences; *For. Sci.* = Formal sciences.

research facilities (such as laboratories, libraries and research sites) emerged as the most frequent (21.9%) among respondents. This issue, although common to many (overall 48.5%) and in all academic fields, was significantly more prevalent among Ph.D. students (66.9%), younger researchers (55.2%) and extremely frequent in the field of Humanities (88.9%). This issue has been widely recognized as a main problem to research activities, especially scientific, during the pandemic. In many cases, the typical worksite has been replaced by working at home in inadequate environmental conditions for the younger population [3]. The impact on clinical science was also significantly affected due to many clinical trials having been paused or terminated due to pandemic restrictions, quarantines and lockdowns; similarly, patient enrolment for new studies has been halted. These circumstances have prompted a specific set of publications from the US Food and Drug Administration [27] and from the European Medical Agency [28] guidelines on the conduct of clinical trials during the COVID-19 pandemic to preserve trial integrity, to ensure compliance and to assure subjects' and patients' safety.

On the other hand, when considering all problems encountered, travel limitations restricting abroad stay, study and/or research periods were felt as the most frequent issue overall (51.3%), despite having been raised as the main issue by only 17.6% of respondents. Ph.D. students, alongside full professors, reported this issue the most (62 and 62.4%, respectively), with

**Table 7. Presence of economic burden, causes and suggestions to improve research according to academic position.**

| | PhD student | Fixed-term researcher | Full-term researcher | Associate professor | Full professor | p |
|---|---|---|---|---|---|---|
| n | 121 | 71 | 71 | 196 | 85 | |
| **Presence of any economic burden (%)** | 15.7 | 15,5 | 22,5 | 23 | 31.8 | 0.05 |
| **List of reasons for economic burden (%)** | | | | | | |
| Unable to allocate funds | 22.2 | 60 | 26.7 | 33.3 | 40.7 | 0.30 |
| Reduction in clinical and scientific activity | 16.7 | 20 | 20 | 30.8 | 44.4 | 0.25 |
| Increased expenses (comprising private costs) | 38.9 | 10 | 40 | 23.1 | 7.4 | **0.044** |
| Interrupted mobility programs | 11.1 | / | / | 10.3 | 3.7 | 0.47 |
| Reduction in the number of calls and grants available | / | 10 | 6.7 | / | 3.7 | 0.35 |
| Closure of archives and libraries | 11.1 | / | / | / | / | **0.034** |
| Lack of personnel | / | / | 6.7 | 2.6 | / | 0.56 |
| **Suggestions to improve academic research (%)** | | | | | | |
| Economic aid | 16.4 | 25 | 25.7 | 23.5 | 24.6 | 0.75 |
| Reduction in bureaucracy | 10.4 | 5.6 | 20 | 29.4 | 22.8 | **0.005** |
| Enhancement of the scientific and clinical activities | 17.9 | 19.4 | 14.3 | 18.6 | 24.6 | 0.80 |
| Increasing the PhD length and ease of access | 31.3 | 11.1 | 2.9 | 5.9 | 1.8 | **<0.001** |
| Digitalization | 4.5 | 13.9 | 17.1 | 9.8 | 8.8 | 0.29 |
| Resumption of presence activities | 4.5 | 5.6 | 8.6 | 5.9 | 8.8 | 0.86 |
| Re-opening of archives and libraries | 10.4 | 11.1 | 2.9 | 2.9 | 3.5 | 0.13 |
| Increasing the personnel | 1.5 | 8.3 | 5.7 | 2 | 1.8 | 0.24 |
| Enhancement of abroad mobility programs | 3 | / | 2.9 | 2 | 3.5 | 0.84 |

Note: significant *P*-values are presented in bold. Academic fields were categorized into the 3 major scientific branches, **Formal sciences** (mathematics, logic, statistics, computer science, etc.), **Natural sciences** (physics, chemistry, biology, earth sciences, etc.) and **Social sciences** (archaeology, geography, anthropology, psychology, sociology, economics, political science, cultural/ethnic studies, gender studies, etc.) and **Humanities** (literature, philosophy, theology, linguistics, human history, arts, etc.) and **Professions and applied sciences** (medicine, law, agriculture, education, physical performance, consumer science, journalism, military sciences, engineering, architecture and design, transportation, business, etc.). Abbreviations: **PAS** = Professions and Applied Sciences; **Hum** = Humanities; **NS** = Natural Sciences; **Soc. Sci.** = Social Sciences; **For. Sci.** = Formal sciences.

**Table 8. Logistic regression analysis of the presence of economic burden.**

| | b | SE | β | p |
|---|---|---|---|---|
| **Presence of economic burden** | | | | |
| Constant | -0.93 (-1.99, 0.03) | 0.56 | | 0.99 |
| Age, years | -0.02 (-0.05, 0.01) | 0.02 | 0.98 | 0.21 |
| Sex, male | -0.30 (-0.74, 0.16) | 0.23 | 0.74 | 0.18 |
| Fixed-term researcher | 0.17 (-0.82, 1.04) | 0.47 | 1.18 | 0.70 |
| Full-term researcher | 0.78 (-0.26, 1.81) | 0.52 | 2.19 | 0.11 |
| Associate professor | 0.87 (-0.13, 1.91) | 0.47 | 2.38 | 0.07 |
| Full professor | 1.36 (0.11, 2.62) | 0.57 | 3.89 | **0.016** |

*Note*: 95% bias corrected accelerated (BCa) confidence intervals, SE and *P* values from 2000 bootstrapped samples are reported in parentheses. Significant *P*-values are presented in bold. *P* for the model = 0.06; $R^2_N$ = 0.034.

Abbreviations: **β** = standardized coefficient; **b** = unstandardized coefficient; **SE** = standarderror

the highest rates in the fields of Humanities (69.4%) and Formal Sciences (56.8%). This limitation has also affected scientific exchange, which has produced a global shift of many international conferences into virtual platforms for teaching and knowledge dissemination, thus somewhat limiting the opportunities for networking [17] and producing effects and "technostress" burnout [18]. On the other hand, this is associated with relevant advantages such as reduced cost for trips and meetings [29]. A decrease in scientific in-person contacts and networking during the pandemic [29], with subsequent impaired opportunities for interactions and peer support, has been reported [1]. Contrarily, global scientific collaboration has reached unprecedented levels, owing to data sharing about COVID-19 related analyses. Although fruitful, such endeavors have raised numerous issues including how to uphold the basic standards of scientific conduct and integrity during the pandemic [12, 30, 31] Our survey also focused on the incidence of pandemic-associated economic burden on academics. Some degree of burden was reported by approximately 1 out of 5 respondents, with no significant differences according to gender, age or academic field. Inability to allocate research funds and a reduction in clinical activities were the most common reasons behind this burden. Indeed, with the prioritization of COVID-19 studies, new clinical trials have been paused, with detrimental impacts on clinical academics involved in non-COVID 19 research [32]. However, increased expenses were also significantly present among Ph.D. students and researchers, especially in the field of Humanities. In a multiple logistic regression analysis to identify the predictors of an economic burden, only the role of Full professor was independently associated with (any) economic burden, possibly because of the larger amount of funds managed. Part of our findings are in accordance with published literature, which reports a higher impact on Early Researcher Career (ERC) and on "wet laboratory researchers" [33, 34]. As PhD students' and fixed-term researchers' positions are usually based on fellowship programs lasting a limited amount of time, the lost research time has reasonably determined a heavy toll on these researchers, as they are expected to produce their scientific work and reach their research objectives within a limited time span [35]. Consequently, this academic population has been particularly vulnerable to the impact of these conditions on their research output, because of delays in publication, partly deriving from difficulties in completing experimental work. All these factors may reduce opportunities for ERCs funding applications and for job applications, in the current context of job cuts and hiring freeze due to global financial pressures. On the other hand, many journals have granted deadline extensions for submitting academic work, financial reporting of ongoing projects has been postponed by some research institutions and universities, and the time limit for the expense of funding derived from grants has often been extended [36, 37]. Wet researchers were particularly affected from the closure of laboratories and research institutions, as the loss of experimental work has been estimated to be between 1 and 6 months in approximately 25% of cases, compared to "dry research" that can be at least in part conducted remotely [2]. Specifically, as previously reported, researchers working with cell cultures suffered widespread supply shortages concerning reagents and consumables (especially plastic ware), but also including personal protective equipment (PPE), such as gloves and masks [3, 7]. Similarly, researchers working with laboratory animals also experienced difficulties deriving from staff shortages, disruption in the supply chain of drugs, necessary tools for animal care and housing [2, 7]. Our results partly differ from the available literature. No significant differences were found in the rates of research-related issues among male and female researchers, except for greater difficulties for women concerning patients' enrolment and access, possibly due to a higher number of female physicians in our sample. Regarding researchers experiencing an economic burden, no significant differences were found between males and females, with a trend favoring females.

An equal rate of men and women experiencing difficulties with family organization impacting their research activities emerged from our survey (25.8% vs 27.1%, respectively). This contrasts with published literature and reports focusing on the risks intrinsic to gender inequality in the academic context. In fact, female academics have been reported to be more likely to assume parenting or domestic responsibilities than their male counterparts [38]. Furthermore, female researchers are more likely to shoulder domestic duties following the closure of universities and research institutions, as well as childbearing and home-schooling when lockdowns also involved schools and kindergartens [39]. Indeed, it has been postulated that many women are currently doing 'second and third shifts' regarding housework and welfare after their first shift of paid work [40–43]. The possible explanation of our data relies on the lack of information on family composition and the presence of offspring from participants in our survey, as parents are expected to suffer more significantly from the issues mentioned above [3]. Moreover, we do not have data about how the number of working hours has changed during the pandemic for male and female researchers. Indeed, data suggest a differential amount of available working hours for male/female and parent/not-parent researchers [3]. Recent evidence suggest that gender-related issues emerged during COVID-19 have lasted well beyond the first wave. In 2020–2021 gender inequalities for women working in biology, biochemistry, and civil and environmental engineering at universities have remained steady or worsened [44]. Compared to men, a greater proportion of women who responded to surveys reported inability to focus on their research activities. Moreover, female scientists were also more likely to report having grant disruptions and research grants that face financial difficulties due to the pandemic [45]. Furthermore, women make up only a third of authors named on COVID-19 related publications, and they are particularly underrepresented among first and last authorship positions [46]. An analysis on more than 2000 Elsevier journals during the first wave of the pandemic has shown that women submitted proportionally fewer manuscripts than men especially as the first author. This deficit was especially pronounced among junior cohorts of women academics [47]. This gender-related issue requires more extensive analysis, since our current survey may not produce sufficient information. As our work aimed at identifying key issues and possible strategies that could help resume academic activities after the pandemic, we asked our community for suggestions. Notably, economic aid was felt necessary by 22.6%, especially by respondents in the field of Natural Sciences (38.5%). In contrast, older researchers favored a reduction in bureaucracy in approximately a quarter of cases, Ph.D. students unsurprisingly suggested to increase the Ph.D. length and ease of access (31.3%) and Humanities researchers supported the need for a more widespread process of digitalization (22.4%) and advocated for the re-opening of archives and libraries (15.8%).

## Limitations

The present survey consisted of closed-ended and open-ended questions sent by email to people involved in research activities in *Sapienza University of Rome*. Some elements were not intended to be investigated in the survey, such as the quantification of the amount of time dedicated to research and different activities (teaching, mentoring, etc.) during the pandemic as well as the quality of remote work from home. Furthermore, our approach was focused on the subjective perceptions of the impact and consequences of COVID-19 on their own research, in qualitative rather than quantitative terms. Moreover, the composition of the family of respondents was not assessed, thus limiting the chance to derive conclusions about issues specifically related to parenthood. Lastly, it is unclear whether the findings and the conclusions drawn from our study may be similarly applied to other academic institutions in Italy or in Europe. Although our study does not reflect a multicenter approach, it has been conducted in the

largest Italian University and one of the largest academic communities in Europe and its findings are largely consistent with previous similar studies.

## Conclusions

To our knowledge, this is the first thorough exploratory analysis of the impact of the COVID-19 pandemic on a population sample such as that included in our survey, which represents the whole active academic multidisciplinary research community in a large public University. This provides many new insights for planning systematic changes to scientific institutions before and during new possible global crises. We had a heterogeneous study population in terms of position, rank, career stage, tenure status, discipline, and characteristics of work. Based on our data, the COVID-19 pandemic caused critical issues at each level of the academic pyramid, from doctoral students to full professors. In fact, COVID-19 had severe consequences on access to facilities and on research grantees, reduced opportunities for international mobility, and exacerbated pre-existing problems such as difficulty in securing research funding. Our results may represent a significant reference for policy makers, university administrators, and scientists about issues exacerbated by COVID-19 related policies and suggest a fundamental need to develop approaches for future crises. Indeed, our findings, beside their potential intrinsic informative value, may be useful to design appropriate policies aimed at limiting the impact of unexpected catastrophic events on research, pointing out what could be needed to preserve and even reinforce research facilities and activities in critical situations like those caused by the SARS-COV-2 pandemic.

## Supporting information

**S1 Appendix. Questionnaire in English.**
(DOCX)

**S2 Appendix. Checklist for Reporting Of Survey Studies (CROSS).**
(DOCX)

## Acknowledgments

The authors are grateful to the members of Sapienza Governance Research Team: Mariasabrina Sarto, Francesca Bozzano, Carlo Catalano, Luciano Galantini, Marco Oliverio, Daniele Riccioni, Mauro Valorani, Antonella Cammisa, Ciro Franco, Andrea Riccio, Fabio Sciarrino.

Francesco Carlomagno, M.D., Ph.D designed and conducted the statistical analysis, generated data tables, interpreted the results, wrote the first draft of the manuscript and was involved in the revision process. Allegra Battistoni M.D., Ph.D wrote the first draft of the manuscript and was involved in the revision process.

Melwyn Luis Muthukkattil, native English language speaker, has revised the manuscript.

## Author Contributions

**Conceptualization:** Massimo Volpe, Massimo Ralli, Andrea Isidori.

**Data curation:** Massimo Volpe, Massimo Ralli, Andrea Isidori.

**Formal analysis:** Massimo Ralli, Andrea Isidori.

**Investigation:** Massimo Volpe, Massimo Ralli, Andrea Isidori.

**Methodology:** Massimo Volpe.

**Supervision:** Massimo Volpe.

**Writing – original draft:** Andrea Isidori.

**Writing – review & editing:** Massimo Volpe, Massimo Ralli.

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
