## [Decision Letter · Decision Letter 0]

2 May 2023

PONE-D-22-11862“The impact of the COVID-19 pandemic on the largest Italian academic community: "Sapienza" University of Rome”PLOS ONE

Dear Dr. Volpe,

Thank you for submitting your manuscript to PLOS ONE. After careful consideration, we feel that it has merit but does not fully meet PLOS ONE’s publication criteria as it currently stands. Therefore, we invite you to submit a revised version of the manuscript that addresses the points raised during the review process. First of all, apologies for the delay. The number of reviewers available for assessing this specific submission was little, and it was really difficult to achieve a second referee to provide an informed report on it. Once seen the review reports, my impression is that the paper needs several improvements and a further review to validate these amendments. Please consider (and respond to) the whole set of comments appended in this communication into the rebuttal letter of your paper, if you decide to revise it.

We look forward to receiving your revised manuscript.

Kind regards,

Sergio A. Useche, Ph.D.

Academic Editor

PLOS ONE

4. Check the last line of the abstract to ensure it is the same.  The reason for this check is to ensure that the AEs and Reviewers are sent correct information to allow them to make a good decision on whether they can manage/review the manuscript.  Only send back for a change if the abstract on EM and in the manuscript are VASTLY different.

Reviewers' comments:

Reviewer's Responses to Questions

**Comments to the Author**

1. Is the manuscript technically sound, and do the data support the conclusions?

Reviewer #1: Partly

Reviewer #2: Partly

2. Has the statistical analysis been performed appropriately and rigorously? 

Reviewer #1: Yes

Reviewer #2: No

3. Have the authors made all data underlying the findings in their manuscript fully available?

Reviewer #1: No

Reviewer #2: No

4. Is the manuscript presented in an intelligible fashion and written in standard English?

Reviewer #1: No

Reviewer #2: No

5. Review Comments to the Author

Reviewer #1: Abstract;

Needs some review

Introduction:

The authors proposed an interesting research, however the introduction needs to some adjustments.

What is the context of the study (needs improvement) ; explore the importance of the theme (needs improvement) ; the main goals and relevance of the study (good) how this study will contribute ( add).

L60-61 discuss better: how double/triple journey of female researcher would be impacted specific in the context of the pandemic? The sentence is very general and accounts for what is already known. Ref.:

Davis, P.B., Meagher, E.A., Pomeroy, C. et al. Pandemic-related barriers to the success of women in research: a framework for action. Nat Med 28, 436–438 (2022). https://doi.org/10.1038/s41591-022-01692-8

L62-64 what about the race to research and publish articles in health science?Investiments in vaccines? Data sharing and cooperation internationally?

Methods

Why you don't have ethical approval for this research?

How researchers developed the informed consent for participants? how you developed the questionnaires? you should at least get a weaver from the ethical committee of the university. Please contact the ethical committee of your institution and clarify this.

Results

Table 1

research type - check "filed"is it a research type? I don't understand.

Supplementary data

make sure to add data that are relevant to the study. Check the journal policy regarding data sharing and transparency in order to be in accordance to it

Conclusions

Make sure to add a bullet point

Limitations

It is always interesting to point it out

Reviewer #2: The authors attempted to assess the effects of the COVID-19 pandemic on research activities of a univeristy in Italy and identify the most critical issues, from a cross-sectional study. The findings may provide some evidence of the pandemic impacts on the research community in an insitution.

The manuscript requires some substantial revisions before consideration of the journal:

1. the literature review was not organised and presented with sufficient evidence

2. the statistical analysis part was not elaborated and explained clealry, especially how the criteria and assumptions of a partciular statistical analysis were met, e.g. some questions in the survey allow selecting multiple answers, how did these considered in the statistical tests

3. the survey adopted was self-designed, some elaboration of the evidence of referencing to literature is required

4. the discussion part appears lacking in-depth discussion, some substantiation of the literature is required

5. an Italian academic insitution was selected as the study site, given its specificity in its organisation/system/roles of academic personnel as stated by the authors , the authors are required to add to explain how the findings would be useful to the wider academic community in other countries

6. PLOS authors have the option to publish the peer review history of their article (what does this mean?). If published, this will include your full peer review and any attached files.

Reviewer #1: **Yes: **Rafaelly Stavale

Reviewer #2: No

---

## [Author Response · Author response to Decision Letter 0]

27 Jun 2023

Answer: done

Response: done

Response: Following journal’s requirements, we have uploaded the full anonymized dataset underlying the findings described in this manuscript in Figshare. DOI to uploaded file is:

10.6084/m9.figshare.23551743 We have integrated the materials and methods section specifying that the full dataset has been uploaded in Figshare. Data Availability Statement: Data is available from Figshare (DOI: 10.6084/m9.figshare.23551743).

4. Check the last line of the abstract to ensure it is the same.  The reason for this check is to ensure that the AEs and Reviewers are sent correct information to allow them to make a good decision on whether they can manage/review the manuscript.  Only send back for a change if the abstract on EM and in the manuscript are VASTLY different.

Response: Dear Editor, the abstract has been changed. 

Reviewers' comments:

Reviewer's Responses to Questions

Comments to the Author

1. Is the manuscript technically sound, and do the data support the conclusions?

Reviewer #1: Partly

Reviewer #2: Partly

2. Has the statistical analysis been performed appropriately and rigorously? 

Reviewer #1: Yes

Reviewer #2: No

3. Have the authors made all data underlying the findings in their manuscript fully available?

Reviewer #1: No

Reviewer #2: No

4. Is the manuscript presented in an intelligible fashion and written in standard English?

Reviewer #1: No

Reviewer #2: No

5. Review Comments to the Author

Reviewer #1: Abstract;

Needs some review

Introduction:

The authors proposed an interesting research, however the introduction needs to some adjustments. What is the context of the study (needs improvement); explore the importance of the theme (needs improvement); the main goals and relevance of the study (good) how this study will contribute (add).

Response: We thank the reviewer for this comment. We have largely modified the introduction. In particular, we added in the introduction section paragraphs giving more insights and context in terms of existing literature, the importance of the subject matter, and the relative contribution of our study to the scientific knowledge that is currently available. Please see the red parts in the text.

L60-61 discuss better: how double/triple journey of female researcher would be impacted specific in the context of the pandemic? The sentence is very general and accounts for what is already known. Ref.: Davis, P.B., Meagher, E.A., Pomeroy, C. et al. Pandemic-related barriers to the success of women in research: a framework for action. Nat Med 28, 436–438 (2022). https://doi.org/10.1038/s41591-022-01692-8

Response: We thank the reviewer for this comment. Gender-related issues in research and teaching during the pandemic have been largely addressed in the new version with the relevant references. Generally, it seems that pre-existing barriers to academic activity have become amplified during the COVID-19 pandemic. Our results are not fully consistent with some of the previous reports, probably as a consequence of some features of the information required in the questionnaires which we acknowledged as a limitation. We have therefore added a paragraph about female researchers facing the pandemic in the introduction as well as a specific discussion. Please see red parts in the text. 

L62-64 what about the race to research and publish articles in health science? Investments in vaccines? Data sharing and cooperation internationally?

Response: We thank the reviewer for this comment. This is a complex issue. During the pandemic, research activities related to Covid-19 have been privileged with a consequent increase in publication of COVID-19 related papers also owing to international unpreceded data sharing. Despite this, a reduction in international cooperation and and meetings in other fields has been observed and the accuracy of fast-track COVID-19 related publications has also been questioned. We have added multiple sentences in both the introduction and discussion about this. Please see red text in the manuscript (see introduction line 71-75; discussion page 26). This aspect was not an objective of our survey although it has been extensively investigated in previous studies as we mentioned in the introduction and discussion.

Methods

Why you don't have ethical approval for this research?

How researchers developed the informed consent for participants? how you developed the questionnaires? you should at least get a weaver from the ethical committee of the university. Please contact the ethical committee of your institution and clarify this.

Response: please see method section for the ethical approval.

Results

Table 1

research type - check "filed"is it a research type? I don't understand.

Response: We are afraid the Reviewer may have misread “field (research)” as “filed (research)”. We intend Field research (or study) to comprise all subjects in which raw data is collected outside of a laboratory, workplace or library setting. This large field comprises social sciences observing people interact in their natural environment, biologists studying natural animal behaviour, and many archaeologists, anthropologists, economists, etc.

Supplementary data

make sure to add data that are relevant to the study. Check the journal policy regarding data sharing and transparency in order to be in accordance to it

Response: We thank the Reviewer for his/her correct observation. We now provide the complete anonymized Excel database used for the current research and all the statistical analyses.

Conclusions

Make sure to add a bullet point

Response: We thank the reviewer for this comment. We don’t know if a bullet point it is appropriate as for journal style. If so, we have provided a separate file with the bullet list.

Limitations

It is always interesting to point it out

Response: We thank the Reviewer for this comment. We added a Limitations section.

Reviewer #2: The authors attempted to assess the effects of the COVID-19 pandemic on research activities of a univeristy in Italy and identify the most critical issues, from a cross-sectional study. The findings may provide some evidence of the pandemic impacts on the research community in an insitution.

The manuscript requires some substantial revisions before consideration of the journal:

1. the literature review was not organised and presented with sufficient evidence

Response: we thank this reviewer for the comment. We have largely integrated both introduction and discussion with the available pertinent literature. Please see red text.

2. the statistical analysis part was not elaborated and explained clealry, especially how the criteria and assumptions of a partciular statistical analysis were met, e.g. some questions in the survey allow selecting multiple answers, how did these considered in the statistical tests

R. We thank Reviewer #2 for his considerations and constructive criticism. Survey questions that allowed multiple answers were analysed and presented as follows: 1) in Table 2 with regards to the description of the research-related issues (no statistical testing); 2) in Tables 3 through 4 with regards to the overall research-related issues experienced based on academic position and according to the impact of gender, age, academic field (using ANOVAs, with no post-hoc testing); 3) in Tables 5 and 8 to assess independent predictors of the overall number of research related issues and the presence of economic burden (using multiple linear regression); 4) in Tables 6 and 7 to describe and assess the reasons for economic burden and suggestions to improve academic research, according to above-mentioned subgroups (using ANOVAs, with no post-hoc testing).

Given the exploratory nature of our study, in which data are collected with an objective (“to explore the impact of the COVID-19 pandemic on research activities and in the attempt to identify the most critical issues”), rather than with prespecified key hypotheses, we chose not to adjust for multiple comparisons, according to relevant literature on the topic (10.1530/EJE-20-1375, 10.1016/j.joca.2016.01.008, 10.1016/S0895-4356(00)00314-0). As such we place low emphasis on specific significant comparisons, but rather focus on the description of the research cohort and on exploring risk factors for greater research burden. Lastly, we employed bootstrapping on 2000 samples as a robust regression methods to account for non-normal data distribution, reporting bias-corrected accelerated (BCa) 95% confidence intervals and adjusted R2 values (according to Nagelkerke) to describe the fit of our models. We have made these points clearer to the reader in the manuscript.

3. the survey adopted was self-designed, some elaboration of the evidence of referencing to literature is required

Response: we thank the Reviewer for his/her correct observation. The questionnaire was indeed self-designed from the authors, who serve(d) as Governance Research Team at Sapienza University of Rome, based on the available literature on the topic, referenced in the Introduction and the Discussion sections. Specifically, the development of the survey focused on exploring the differential impact of the pandemic on different academic subgroups, expecting a greater toll on women (PMID 32409466), younger researchers and Ph.D. students (PMID 32493949), specific research activities (PMID 36530543), and researchers involved in clinical trials (35050229), etc. All these aspects are now better discussed in the Introduction section and the methodology is more clearly presented in Lines 149-151.

4. the discussion part appears lacking in-depth discussion, some substantiation of the literature is required

Response: we thank this reviewer for the comment. We have largely integrated both introduction and discussion with available literature. Please see red text.

5. an Italian academic institution was selected as the study site, given its specificity in its organization/system/roles of academic personnel as stated by the authors, the authors are required to add to explain how the findings would be useful to the wider academic community in other countries

Response: we thank this reviewer for the comment. As now clearly stated in the text, Sapienza is the largest university in Italy with many excellent departments. Academics of all degrees have taken part to this survey, therefore the authors think that results could be useful to a wider academic community. In order to meet this appropriate remark of the reviewer, we also proposed a modified title of the article which reflects the impact of pandemic of a very representative academic research community and on potential other institutions.

---

## [Decision Letter · Decision Letter 1]

15 Jan 2024

PONE-D-22-11862R1“The impact of the COVID-19 pandemic on research activities in the largest Italian academic community”PLOS ONE

Dear Dr. Volpe,

Thank you for submitting your manuscript to PLOS ONE. After careful consideration, we feel that it has merit but does not fully meet PLOS ONE’s publication criteria as it currently stands. Therefore, we invite you to submit a revised version of the manuscript that addresses the points raised during the review process. After a careful re-review, our Referee has changed their mind and suggests allowing you to conduct a further set of amendments on the basis of the comments appended below (see attached comments). Although the paper has not been advised for rejection anymore, but for major revisions, please perform a rigorous set of adjustments, given that the potential acceptance of it depends on the approval of the Reviewer in consideration of your amendments.

We look forward to receiving your revised manuscript.

Kind regards,

Sergio A. Useche, Ph.D.

Academic Editor

PLOS ONE

Reviewers' comments:

Reviewer's Responses to Questions

**Comments to the Author**

1. If the authors have adequately addressed your comments raised in a previous round of review and you feel that this manuscript is now acceptable for publication, you may indicate that here to bypass the “Comments to the Author” section, enter your conflict of interest statement in the “Confidential to Editor” section, and submit your "Accept" recommendation.

Reviewer #1: (No Response)

2. Is the manuscript technically sound, and do the data support the conclusions?

Reviewer #1: Partly

3. Has the statistical analysis been performed appropriately and rigorously? 

Reviewer #1: No

4. Have the authors made all data underlying the findings in their manuscript fully available?

Reviewer #1: No

5. Is the manuscript presented in an intelligible fashion and written in standard English?

Reviewer #1: No

6. Review Comments to the Author

Reviewer #1: Dear authors,

The manuscript is about an important topic and at several moments you addressed it well. However I have a few suggestions that may improve your research report.

Study report and methods:

please check the two references bellow about survey report. They are about the checklists you can use to increase quality and transparency of your research report. I suggest to use it, add to you methods section and to the supplementary files the checklists done.

Eysenbach G. Improving the quality of Web surveys: the Checklist for Reporting Results of Internet E-Surveys 1. (CHERRIES). J Med Internet Res. 2004 Sep 29;6(3):e34. doi: 10.2196/jmir.6.3.e34. Erratum in: doi:10.2196/jmir.2042. PMID: 15471760; PMCID: PMC1550605.

2. Sharma A, Minh Duc NT, Luu Lam Thang T, Nam NH, Ng SJ, Abbas KS, Huy NT, Marušić A, Paul CL, Kwok J, Karbwang J, de Waure C, Drummond FJ, Kizawa Y, Taal E, Vermeulen J, Lee GHM, Gyedu A, To KG, Verra ML, Jacqz-Aigrain ÉM, Leclercq WKG, Salminen ST, Sherbourne CD, Mintzes B, Lozano S, Tran US, Matsui M, Karamouzian M. A Consensus-Based Checklist for Reporting of Survey Studies (CROSS). J Gen Intern Med. 2021 Oct;36(10):3179-3187. doi: 10.1007/s11606-021-06737-1. Epub 2021 Apr 22. PMID: 33886027; PMCID: PMC8481359.

The authors approach several important topics such as gender inequalities and disparities through science. However it is important to review some of the writing to avoid misleading conceptions. I suggest to improve the references used throughout the manuscript and to be careful when it comes to date of publications as well. Specially because early COVID-19 studies didn't had sufficient data.

Introduction:

The authors already worked on the introduction, still there are still some tings that should be taken into account.

Why the study is important? What it can add to the scientific knowledge? (observations at the file attached).

Methods:

Please explain more about how the survey was developed. Use the checklists suggested above.

Regarding the survey - what the authors mean about "family organization"?

Regarding the "economic burden" - did you asked about it in the previous year? since the beginning of the pandemic? Did you consider a burden as a reduction at family income or just the respondent income?

Have you performed a pilot study to test the survey?

Stats: please add the tests results as supplementary information. the results of the stats tests will assist others to better understand your results and will increase transparency.

Bootstrapping is not commonly used. Have you tested for normality before using bootstrapping ? The test may result in a few limitations for stats inference that was not addressed by the authors on the limitation section.

More comments are at the file attached.

Results:

It would be interesting to add a flowchart of the survey production and results of respondents - at the beginning of the results section.

Please check the references I add previously about survey reports.

Discussion:

It requires a few adjustments to correlates the authors findings with the current literature. Use this section to add how your main findings can be useful for future policies and practices in science.

Please check the file attached with more observations.

Conclusion:

What are the main findings of your study? And a final recommendation?

The conclusion is very broad and could reinforce the main findings and how it impacts science.

OBS: It is not because the study was conducted in Italy that its findings are restricted to it or European countries. Science is borderless, find a way to express the general importance of your findings.

References

Please make sure to cite the papers at the end of each sentence;

some references are probably in the wrong place - please check the file attached.

Add more relevant references to support you idea. I suggested a few that you are not obligated to use.

7. PLOS authors have the option to publish the peer review history of their article (what does this mean?). If published, this will include your full peer review and any attached files.

Reviewer #1: **Yes: **Rafaelly Stavale

---

## [Author Response · Author response to Decision Letter 1]

25 Feb 2024

Point-by-point Responses to Comments to the Authors

Reviewer #1: Dear authors,

The manuscript is about an important topic and at several moments you addressed it well. However, I have a few suggestions that may improve your research report.

Study report and methods:

please check the two references bellow about survey report. They are about the checklists you can use to increase quality and transparency of your research report. I suggest to use it, add to your methods section and to the supplementary files the checklists done.

Eysenbach G. Improving the quality of Web surveys: the Checklist for Reporting Results of Internet E-Surveys 1. (CHERRIES). J Med Internet Res. 2004 Sep 29;6(3):e34. doi: 10.2196/jmir.6.3.e34. Erratum in: doi:10.2196/jmir.2042. PMID: 15471760; PMCID: PMC1550605.

2. Sharma A, Minh Duc NT, Luu Lam Thang T, Nam NH, Ng SJ, Abbas KS, Huy NT, Marušić A, Paul CL, Kwok J, Karbwang J, de Waure C, Drummond FJ, Kizawa Y, Taal E, Vermeulen J, Lee GHM, Gyedu A, To KG, Verra ML, Jacqz-Aigrain ÉM, Leclercq WKG, Salminen ST, Sherbourne CD, Mintzes B, Lozano S, Tran US, Matsui M, Karamouzian M. A Consensus-Based Checklist for Reporting of Survey Studies (CROSS). J Gen Intern Med. 2021 Oct;36(10):3179-3187. doi: 10.1007/s11606-021-06737-1. Epub 2021 Apr 22. PMID: 33886027; PMCID: PMC8481359. 

R. We thank the Reviewer for her detailed and constructive comments and the suggestion to include the use of a survey checklist. We adopted the CROSS checklist by Sharma A. et al. and we present it in Supplementary material as Appendix S2. We further describe its use in the Methods section [Lines 149-151].

The authors approach several important topics such as gender inequalities and disparities through science. However, it is important to review some of the writing to avoid misleading conceptions. I suggest to improve the references used throughout the manuscript and to be careful when it comes to date of publications as well. Specially because early COVID-19 studies didn't have sufficient data. 

R. We thank the Reviewer for this comment. We have revised the whole manuscript with the support of an English mother-tongue person. In addition, we checked the references throughout the manuscript and we added new refs as suggested or appropriate. 

Introduction:

The authors already worked on the introduction, still there are still some things that should be taken into account.

Why the study is important? What it can add to the scientific knowledge? (observations at the file attached). 

We thank the reviewer for this comment. We have acknowledged all observations related to the Introduction in the manuscript. See text .

Methods:

Please explain more about how the survey was developed. Use the checklists suggested above.

Regarding the survey - what the authors mean about “family organization”?

Regarding the “economic burden” - did you asked about it in the previous year? Since the beginning of the pandemic? Did you consider a burden as a reduction at family income or just the respondent income?

R. We thank the Reviewer for her observations. For the use of checklists see the point mentioned above. The questionnaire proposed to the participants was referred to the period going from the onset of the pandemic, up to April 2021. As it can be seen from the online survey administered to all participants (provided in full in the Supplementary material), the terms “family organization” (to make it more clear we now define this field as “impact on family organization”), “economic burden”, etc. were indeed chosen by the participants in their responses. As such, no specific definition or interpretation can be given by the authors; this is what the respondents chose the answer(s) as they felt it appropriate on the basis of their subjective experience. We agree with the Reviewer that it would be interesting to have more insights on the impact of the pandemic on “family problems” or “economic burden”, but this should be addressed in specifically designed studies. 

Have you performed a pilot study to test the survey?

R. We did not conduct a pilot study. This is an obvious consequence of the sudden onset of the pandemic which prevented any previous experience.

Stats: please add the tests results as supplementary information. the results of the stats tests will assist others to better understand your results and will increase transparency.

Bootstrapping is not commonly used. Have you tested for normality before using bootstrapping? The test may result in a few limitations for stats inference that was not addressed by the authors on the limitation section.

More comments are at the file attached. 

R. We thank the Reviewer for her observations. We would like to observe that the complete dataset was already made publicly available using the data repository Figshare (see point above) (we checked it again and it is fully accessible) in the previous step of the revision process, and we fully describe in the Methods section the complete statistical analysis employed. Furthermore, the results are completely presented in Tables 1 through 8. As such we believe full transparency is guaranteed, both to the Editor, the Reviewers and the readers.

We indeed did analyze data distribution, and employed a robust and vastly used regression method, namely bootstrapping, to account for non-normal data, as specified in the Methods section:

 “Data distribution was visually inspected by analyzing the respective histograms and normality plots. […] A robust approach using bootstrapping for 2000 samples was employed to account for non-normal data distribution, and bias-corrected accelerated (BCa) 95% confidence intervals were calculated and reported.”

Given the numerosity of the study sample size, we believe no specific limitations to study results and conclusions may arise from the use of bootstrapping.

Results:

It would be interesting to add a flowchart of the survey production and results of respondents - at the beginning of the results section.

Please check the references I add previously about survey reports. 

R. We thank the Reviewer for her suggestion. As specified in the Methods section the online survey employed was collegially developed by the Sapienza University Governance Research Team, after careful review of the available literature on the topic, at the time of study design. Furthermore, considering that the study was cross-sectional in nature, and respondents were only asked to complete the survey at a single time-point, we believe that a flowchart would not add sensible information to the readers. With regard to online surveys checklist, we adopted the CROSS checklist by Sharma A. et al., as specified above.

Discussion:

It requires a few adjustments to correlates the authors findings with the current literature. Use this section to add how your main findings can be useful for future policies and practices in science.

Please check the file attached with more observations. 

R. We thank the reviewer for this comment. We have put our results in the context of the existing available literature and added several new references. We introduced a new paragraph about the relevance of our work (lines …..).

Conclusion:

What are the main findings of your study? And a final recommendation?

The conclusion is very broad and could reinforce the main findings and how it impacts science.

R. We have added a short summary of our findings in the conclusion. 

References

Please make sure to cite the papers at the end of each sentence;

some references are probably in the wrong place - please check the file attached.

Add more relevant references to support you idea. I suggested a few that you are not obligated to use.

R. We thank the reviewer for this comment. We have added new references and mentioned them at the end of each sentence in the text.

---

## [Decision Letter · Decision Letter 2]

17 Apr 2024

PONE-D-22-11862R2“The impact of the COVID-19 pandemic on research activities: a survey of the largest Italian academic community”PLOS ONE

Dear Dr. Volpe,

Thank you for submitting your manuscript to PLOS ONE. After careful consideration, we feel that it has merit but does not fully meet PLOS ONE’s publication criteria as it currently stands. Therefore, we invite you to submit a revised version of the manuscript that addresses the points raised during the review process.

Your revised paper has been re-reviewed. I'm happy to say that it will be considered acceptable for publication following minor revisions. Please proceed to apply the changes requested by the Reviewer # 3, and resubmit. As these amendments are slight, I will assess myself their adequacy, thus omitting a new round of reviews if they are sound enough.

We look forward to receiving your revised manuscript.

Kind regards,

Sergio A. Useche, Ph.D.

Academic Editor

PLOS ONE

Journal Requirements:

Additional Editor Comments (if provided):

Reviewers' comments:

Reviewer's Responses to Questions

**Comments to the Author**

1. If the authors have adequately addressed your comments raised in a previous round of review and you feel that this manuscript is now acceptable for publication, you may indicate that here to bypass the “Comments to the Author” section, enter your conflict of interest statement in the “Confidential to Editor” section, and submit your "Accept" recommendation.

Reviewer #1: All comments have been addressed

2. Is the manuscript technically sound, and do the data support the conclusions?

Reviewer #1: Yes

3. Has the statistical analysis been performed appropriately and rigorously? 

Reviewer #1: Yes

4. Have the authors made all data underlying the findings in their manuscript fully available?

Reviewer #1: Yes

5. Is the manuscript presented in an intelligible fashion and written in standard English?

Reviewer #1: No

6. Review Comments to the Author

Reviewer #1: Dear authors,

there is just a few minor points to be revised. I appreciate all the comments and clarifications on the previous review.

Congratulations on your research report.

"All these factors may reduce opportunities for ERCs funding applications and for job applications, in the current context of job cuts and hiring freeze due to global financial pressures. On the other hand, many journals have granted deadline extensions for submitting academic work, financial reporting of ongoing projects has been postponed by some research institutions and universities, and the time limit for the expense of funding derived from grants has often been extended." - > please make sure to add the reference to support this information.

Please correct:

'Although out study does not reflect a multicenter approach, it has been conducted in the largest Italian University and one of the largest academic communities in Europe and its findings are largely consistent with previous similar studies."to (...) our study (...)

7. PLOS authors have the option to publish the peer review history of their article (what does this mean?). If published, this will include your full peer review and any attached files.

Reviewer #1: **Yes: **Rafaelly Stavale

---

## [Author Response · Author response to Decision Letter 2]

3 May 2024

Reviewer #1: Dear authors,

there is just a few minor points to be revised. I appreciate all the comments and clarifications on the previous review.

Congratulations on your research report.

"All these factors may reduce opportunities for ERCs funding applications and for job applications, in the current context of job cuts and hiring freeze due to global financial pressures. On the other hand, many journals have granted deadline extensions for submitting academic work, financial reporting of ongoing projects has been postponed by some research institutions and universities, and the time limit for the expense of funding derived from grants has often been extended." - > please make sure to add the reference to support this information.

Thank you for your comment. We have added some references to the bibliography (n 38-39)

Please correct:

'Although out study does not reflect a multicenter approach, it has been conducted in the largest Italian University and one of the largest academic communities in Europe and its findings are largely consistent with previous similar studies."to (...) our study (...)

Thank you, we have corrected the typo

---

## [Editor Report · Decision Letter 3]

7 May 2024

“The impact of the COVID-19 pandemic on research activities: a survey of the largest Italian academic community”

PONE-D-22-11862R3

Dear Dr. Volpe,

We’re pleased to inform you that your manuscript has been judged scientifically suitable for publication and will be formally accepted for publication once it meets all outstanding technical requirements.

Kind regards,

Sergio A. Useche, Ph.D.

Academic Editor

PLOS ONE

Additional Editor Comments (optional):

Thanks for the soundness of your amendments and for the promptness of the author responses.

---

## [Editor Report · Acceptance letter]

13 May 2024

PONE-D-22-11862R3 

PLOS ONE

Dear Dr. Volpe, 

I'm pleased to inform you that your manuscript has been deemed suitable for publication in PLOS ONE. Congratulations! Your manuscript is now being handed over to our production team.

Kind regards, 

on behalf of

Dr. Sergio A. Useche 

Academic Editor

PLOS ONE